# Endophytic *Bacillus subtilis* SR22 Triggers Defense Responses in Tomato against Rhizoctonia Root Rot

**DOI:** 10.3390/plants11152051

**Published:** 2022-08-05

**Authors:** Younes M. Rashad, Sara A. Abdalla, Mohamed M. Sleem

**Affiliations:** Plant Protection and Biomolecular Diagnosis Department, Arid Lands Cultivation Research Institute, City of Scientific Research and Technological Applications (SRTA-City), New Borg El-Arab City 21934, Egypt

**Keywords:** JERF3, resistance, *Rhizoctonia solani*, solanum lycopersicum, qRT-PCR

## Abstract

Rhizoctonia root rot is one of the most destructive diseases of tomato and other crops. The biocontrol of plant diseases using endophytic bacteria has gained significant attention due to their distinct advantages compared with the free-living ones, as well as their new unexplored and unique properties. Endophytic *Bacillus subtilis* SR22 represents a promising and more effective biocontrol and growth-promoting agent for tomato plants than the free-living agents, being an ecofriendly and sustainable tool in modern agriculture. In this study, the direct antagonistic activity of *B. subtilis* SR22 was investigated against *Rhizoctonia solani* in vitro. The biocontrol activity of *B. subtilis* SR22 against Rhizoctonia root rot of tomato was also investigated. Effects on the level of the transcriptional expression of defense-related genes, biochemical responses, and the vegetative growth of tomato plants were also studied. The dual culture test showed 51% inhibition in the mycelial growth of *R. solani* due to *B. subtilis* SR22, indicating its potent antagonistic behavior. Using a GC-MS analysis, twenty bioactive compounds were detected to be produced by *B. subtilis* SR22, including chlorogenic acid, pyrrolo [1,2-a]pyrazine-1,4-dione, hexahydro, propyl thioglycolic acid, phthalic acid, and 2,3-butanediol. Under greenhouse conditions, the application of *B. subtilis* SR22 led to a reduction (up to 51%) in Rhizoctonia root rot of tomato. Furthermore, an upregulation in the expression of the responsive factor *JERF3* (10.9-fold) and the defense-related genes *POD* (9.1-fold) and *PR1* (4.5-fold) in tomato plants was recorded due to the application of *B. subtilis* SR22. In addition, this treatment enhanced the total phenolic content (76.8%) and activity of the antioxidant enzymes POD (56%) and PPO (29.2%) in tomato roots, indicating its resistance-inducing effect on tomato plants. Moreover, this treatment enhanced most of the evaluated growth parameters in tomato plants (up to 35%). We can conclude that *B. subtilis* SR22 is a promising biocontrol agent and growth promoter in tomato plants against Rhizoctonia root rot. An evaluation of the formulation and field application of this bio-agent is necessary in future studies.

## 1. Introduction

Tomato (*Solanum lycopersicum* L.), a member of the Solanaceae family, is one of the most globally cultivated and consumed horticultural crops [1]. In 2020, the total world production was estimated at 186.8 million tons from a total harvested area of 5 million ha [2]. Tomato is a nutritious and useful food containing several bioactive chemicals such as carotenoids (primarily lycopene and ß-carotene); ascorbic acid (vitamin C); tocopherols (vitamin E); fibers; vitamin A; and polyphenolic compounds (phenols, flavonoids, and phenolic acids) [1]. However, it is susceptible to several fungal pathogens that affect its production worldwide, causing high economic loss [3]. Among them, *Rhizoctonia solani*, *Fusarium* spp., and *Pythium* spp. can attack tomato plants during their early developmental phases, resulting in seed rot, pre-emergence damping-off, or stem rot in greenhouses and open fields [4].

Rhizoctonia root rot, caused by *R. solani* Kühn, is among the most destructive tomato diseases, causing enormous economic loss up to 50% [5]. This pathogen can persist in the soil as mycelia or sclerotia and within infected plant materials for several years, through which the pathogen can transmit to a new, uninfected area. Moreover, *R. solani* is a nonspecific pathogen, which has a broad host range [6]. Disease symptoms include seed rot, pre- and post-emergence damping off, root rot, stem and crown cancer, dwarfism, and may lead to the complete death of the plant under severe conditions [7]. Different anastomosis groups (AGs) of *R. solani* have been reported to infect tomato plants at varying degrees, including AG-2-1, AG-2-2, AG-3-PT, AG-4-HG-I, and AG-5 [8]. A variety of strategies have been used to manage this disease including chemical, physical, crop rotation, resistant cultivars, and biological control [9,10,11]. Among these alternative techniques, biological control is the most favorable, owing to its safety, eco-friendliness, and efficiency [12]. The biological control of *R. solani* on various crops has been extensively studied [10,11,13]. 

Among the biocontrol agents, numerous species belonging to genus *Bacillus* have been studied against different fungal diseases on different crops [14,15,16]. Their ability to form endospores, stimulate plant defenses, compete for resources and space, and produce potent antibiotics may contribute to their efficient bio-control activity [17]. Endophytic bacteria are non-pathogenic bacteria, which inhabit plant tissues for part of their life cycle without causing any disease symptoms [18]. They have various distinct advantages when compared with free-living bacteria, including colonization of an ecological niche similar to that of vascular plant pathogens; facing less competition with other microorganisms; achieving a sufficient supply of nutrients; less exposure to rhizosphere’s abiotic and antibiotic stress factors; and easy translocation of their bioactive metabolites within the host plant [19]. Recently, endophytes have gained significant attention due to their new unexplored and unique properties, as well as bioactive secondary metabolites that sustain their mutualistic interactions with their hosts [20]. Rajendran and Samiyappan [21] reported around a 64% reduction in Rhizoctonia damping off in cotton when treated with the endophytic *Bacillus* spp. (EPCO102 and EPCO16). In addition, a reduction in the mycelial growth of *R. solani* was also achieved in vitro. Various modes of action that may contribute to the endophyte’s antagonistic activity have been discussed, including competition for space or resources and antibiosis through the formation of antibacterial volatile or non-volatile metabolites [22]. In addition, there is another indirect role that endophytes may play in the bio-control via induction of the host immunity to the invading pathogen by activating multiple signaling pathways [21].

The priming of plant immunity results in the activation of multiple hypersensitivity reactions in the host, such as lignin deposition, accumulation of polyphenolic compounds and phytoalexins, and/or overexpression of multiple defense-related genes [23]. Among the up-regulated genes, advantageous entophyte colonization has shown to activate jasmonate and ethylene responsive factor 3 (*JERF3*), which controls a group of defense-responsive genes via jasmonate/ethylene signaling pathways such as peroxidase gene (*POD*), which encodes the antioxidant peroxidase enzyme, and the antifungal pathogenesis-related protein 1 gene (*PR1*) [10,24]. The objectives of this study were to (1) investigate the antagonistic activity of the endophytic *B. subtilis* against *R. solani* in vitro; (2) identify its produced antifungal secondary metabolites; (3) evaluate its bio-control activity against Rhizoctonia root rot of tomato under greenhouse conditions; (4) investigate its effects on the transcriptional level of the defense-related genes *JERF3*, *PR1*, and *POD* in tomato; (5) investigate its effect on the plant growth and biochemical parameters. Beneficial endophytic bacteria may represent a more effective and sustainable solution for plant diseases due to their unique compatibility with the host plant over the free-living bacteria.

## 2. Results

### 2.1. Molecular Identification and Phylogeny of the Endophytic Bacteria SR22

BLAST search results showed that the nucleotide sequence (1402 bp) of the isolated endophytic bacteria SR22 has 100% similarity with *B. subtilis*, which was deposited in GenBank under accession number (ON624459). The constructed phylogeny tree (Figure 1) showed that *B. subtilis* SR22 was grouped with *B. subtilis*, *B. amyloliquefaciens*, and *B. licheniformis* in a distinct cluster with 87% bootstrap support, while *B. anthracis*, *B. cereus*, *B. thuringiensis* and *B. wiedmannii* were grouped in a separate cluster with 92% bootstrap support.

### 2.2. Antagonism Assay of B. subtilis SR22 against R. solani In Vitro

Antagonistic activity of *B. subtilis* SR22 was investigated against *R. solani* in vitro (Figure 2). The obtained results showed that *R. solani* had a fast growth rate covering the full plate in 3 days. In contrast, a considerable inhibition in its growth (51%) was achieved when it was cultured with *B. subtilis* SR22 in the dual culture plate. This result demonstrates the antifungal potential of *B. subtilis* SR22 against *R. solani*.

### 2.3. Gas Chromatography-Mass Spectrometry (GC-MS)

Twenty secondary metabolites produced by *B. subtilis* SR22 were detected by GC-MS system (Figure 3, and Table 1). The results obtained showed that three substances were found in major ratios including propane, 2-ethoxy-(41.47%); chlorogenic acid (33.87%); and pyrrolo [1,2-a]pyrazine-1,4-dione, hexahydro (13.7%). Four compounds were detected in intermediate ratios including 1-hexadecanol (1.9%); propyl thioglycolic acid (1.5%); 2,3-butanediol (1.09%); and phthalic acid, 3,5-dimethylphenyl 4-isopropyl (1.03%), while the other metabolites were detected in trace ratios. 

### 2.4. Effect on Transcriptional Expression of the Defense-Related Genes

The transcriptional expression levels of the defense-response genes *JERF3*, *POD*, and *PR1* were investigated in the tomato plants infected with Rhizoctonia root rot and/or treated with *B. subtilis* SR22 at 7 dpi (Figure 4). The results from the quantitative real-time PCR (qRT-PCR) revealed that treating the tomato plants with *B. subtilis* SR22 triggered the studied defense-related genes in a pathogen-dependent manner. In this regard, it was found that infection with *R. solani* upregulated the responsive factor *JERF3*, compared to the untreated control, while treating the infected tomato plants with *B. subtilis* SR22 led to the highest gene expression (10.9-fold). For *POD*, the obtained results showed that infection with *R. solani* or treatment with *B. subtilis* SR22 significantly overexpressed *POD* at varying degrees when compared with the untreated control treatment. However, the inducing effect of infection with *R. solani* was higher than that due to treating with *B. subtilis* SR22. The highest *POD* expression was recorded for the dual treatment (infected with *R. solani* and treated with *B. subtilis* SR22), recording 9.1-fold. For *PR1*, infection with *R. solani* upregulated *PR1*, compared with the control treatment. However, the highest *PR1* expression was observed for the infected tomato plants, which were treated with *B. subtilis* SR22, recording 4.5-fold. The triggering effect of *B. subtilis* SR22 on the expression of the studied defense-response genes in a pathogen-dependent manner may indicate its protective role against *R. solani*.

### 2.5. Effect on Disease Incidence and Severity

The biocontrol effect of treating the tomato plants with *B. subtilis* SR22 on the severity and incidence of Rhizoctonia root rot at 45 dpi is presented in Table 2. The results obtained showed that no disease symptoms were detected in the tomato plants, which did not receive *R. solani* inoculum, whether treated with *B. subtilis* SR22 or not. In contrast, the tomato plants that only received *R. solani* inoculum exhibited 100% disease incidence with 65% severity. Treating the infected tomato plants with *B. subtilis* SR22 led to a significant reduction in the disease incidence to 60.3% with a severity of 30%, compared with the untreated infected plants.

### 2.6. Effect on Growth Parameters

The effect of treatment with *B. subtilis* SR22 on the growth of tomato plants infected with Rhizoctonia root rot at 45 dpi is presented in Table 3. The results showed that the infection of tomato plants with Rhizoctonia root rot considerably reduced the lengths of tomato shoot and root, as well as their dry weights. In contrast, treating the tomato plants with *B. subtilis* SR22 significantly enhanced all of the evaluated growth parameters, except the number of leaves, when compared with the untreated, non-infected control plants. At the same time, treating the infected plants with *B. subtilis* SR22 led to an improvement in the shoot and root lengths and dry weights, when compared with the untreated infected plants. These results indicate the growth-promoting effect of *B. subtilis* SR22 on tomato plants, whether infected with *R. solani* or not.

### 2.7. Effect on Phenolic Content and Activity of Antioxidant Enzymes

The mean phenolic content and activity of peroxidase (POD) and polyphenol oxidase (PPO) enzymes in the roots of the tomato plants infected with Rhizoctonia root rot and treated with *B. subtilis* SR22 at 45 dpi are presented in Table 4. Data from the biochemical analyses revealed that infection of the tomato plants with *R. solani* led to a significant increment in their total phenolic content and the activity of POD and PPO, compared with the non-infected control plants. The application of *B. subtilis* SR22 on the tomato plants also induced their phenolic content and increased the activities of both POD and PPO. However, the inducing effect due to infection with *R. solani* was higher than that due to *B. subtilis* SR22, when compared with the untreated, non-infected control treatment. The highest values of these parameters were recorded for the tomato plants infected with *R. solani* and treated with *B. subtilis* SR22, compared with the untreated, non-infected tomato plants.

### 2.8. Pearson Correlation Coefficient between the Studied Variables

The correlation coefficient values between the variables studied in the tomato plants are illustrated in Table 5. The results showed that the shoot length of the tomato plants strongly correlated with the root length and shoot dry weight, recording r = 0.91 and 0.90, respectively, at *p* ≤ 0.001, and moderately correlated with the root dry weight (r = 0.72 at *p* ≤ 0.01). In addition, the root length of tomato exhibited moderate correlation with the shoot dry weight (r = 0.82 at *p* ≤ 0.01) and low correlation with the root dry weight (r = 0.62 at *p* ≤ 0.05). Meanwhile, the shoot dry weight moderately correlated with the root dry weight (r = 0.73 at *p* ≤ 0.01). Furthermore, the total phenolic content strongly correlated with both enzymes peroxidase and polyphenol oxidase (r =0.96 and 0.98, respectively at *p* ≤ 0.001), while both enzymes strongly correlated with each other (r = 0.97 at *p* ≤ 0.001). In contrast, no correlations were recorded between the number of leaves, phenolic compounds, peroxidase and polyphenol oxidase enzymes with the other growth variables.

### 2.9. Principal Component Analysis of Studied Variables

The results from the principal component analysis showed that PC1 was represented by the growth parameters (shoot length, root length, dry weight of shoot, dry weight of root, and number of leaves), while PC2 was represented by phenolic content, peroxidase and poly phenol oxidase enzymes (Figure 5). The four tested treatments were well separated from each other on the diagram, indicating that no similarity was observed between them. The three studied variables (phenolic content, peroxidase and poly phenol oxidase enzymes) were more pronounced for the treatment (P+B) than treatment (P), while the studied growth variables (shoot length, root length, dry weight of shoot, dry weight of root, and number of leaves) were more pronounced for treatment (B) than the control treatment (C). 

## 3. Discussion

Rhizoctonia root rot is one of the most destructive diseases of tomato and other crops, causing enormous plant damage and around 50% reduction in the yield. The biocontrol of plant diseases using endophytic bacteria has gained significant attention due to their distinct advantages compared with the free-living ones, as well as their new unexplored and unique properties [19]. The results obtained in this study indicated an effective antagonistic potential of *B. subtilis* SR22 against *R. solani* in vitro. This result is in accordance with the inhibition in the mycelial growth of *R. solani* due to the endophytes *Bacillus* sp. EPCO102 and *Bacillus* sp. EPCO16, reported by Rajendran and Samiyappan [21]. In this regard, different antagonistic mechanisms can be discussed, including competition for space and/or nutrients, in addition to antibiosis via the production of a diverse set of volatile and non-volatile antifungal secondary metabolites, and/or antifungal enzymes [22]. Moreover, around 200 antibiotics with variable structures and potentialities have been reported to be produced by *Bacillus* spp. [25]. The obtained inhibition in the mycelial growth of *R. solani* can be attributed to the competition or production of antifungal metabolites by *B. subtilis* SR22. However, the inhibition zone due to competition for nutrients cannot be visually distinguished from that due to antibiosis. Therefore, we performed a GC-MS analysis for the produced metabolites in the filtered culture of *B. subtilis* SR22 to identify the antifungal constituents. The results from the GC-MS analysis revealed the production of various compounds with an antifungal background, to which the antagonistic behavior of *B. subtilis* SR22 can be attributed, such as chlorogenic acid, pyrrolo [1,2-a]pyrazine-1,4-dione, hexahydro, propyl thioglycolic acid, and phthalic acid. Chlorogenic acid, the main produced metabolite by *B. subtilis* SR22 (33%), has been reported as a potent antifungal compound against different phytopathogenic fungi, inducing lysis of their cells via interfering with their cell permeability [26]. Pyrroles derivatives have been also reported to exhibit antifungal activity via the downregulation of some functional proteins-encoding genes, inhibiting the synthesis of these vital proteins [27]. All of these antifungal metabolites synergistically contributed to the antagonistic behavior of *B. subtilis* SR22.

One of the most interesting results obtained in this study is the effective biocontrol activity of *B. subtilis* SR22 when applied on tomato plants against infection with Rhizoctonia root rot. This result is in agreement with that obtained by Wu et al. [28] for the application of *B. subtilis* SL-44 against pepper seedling wilt, caused by *R. solani*. In addition to the direct antifungal activity of *B. subtilis* SR22 against *R. solani*, our study showed an overexpressing effect of its use on the responsive factor *JERF3* and some defense-related genes in tomato roots (*POD* and *PR1*). *JERF3* regulates many defense genes via jasmonic acid and ethylene signaling pathways against various stresses that affect the plant [24]. In addition, *POD* and *PR1* were also upregulated due to *B. subtilis* SR22. The *POD* gene that encodes for the antioxidant enzyme peroxidase acts as a scavenger of reactive oxygen species (ROS) and is produced due to the infection stress that destroys the plant cell structure, in addition to its role in the detoxification of hydrogen peroxide (H_2_O_2_) [29]. This triggering effect was supported by the elevated activity of POD and PPO enzymes in tomato roots, which was recorded in this study due to *B. subtilis* SR22 application. The recorded overexpression of *PR1*, which encodes for an antifungal pathogenesis-related protein 1, is a further evidence that has long been used as a marker for induced plant immunity [30]. Furthermore, the recorded increment in the phenolic compounds in tomato roots, which have fungitoxic effects on different pathogenic fungi, represents another triggered plant defense mechanism against *R. solani*. In addition to their direct inhibitory effect on the mycelial growth of the fungal pathogen, the deposition of some phenolic compounds such as lignin in the plant cell wall acts as a physical barrier restricting the pathogen transfer from cell to cell [31]. All of these defense mechanisms synergistically contributed to the biocontrol activity of *B. subtilis* SR22 and indicated their roles in triggering tomato resistance against Rhizoctonia root rot. In this regard, the results from the GC-MS analysis revealed production of the bioactive compound 2,3-butanediol by *B. subtilis* SR22. This compound has been widely reported as a resistance inducer for many plants through the ethylene signaling pathway via ROS homeostasis and upregulating the defense-related PR genes [32].

One of the interesting results obtained from this study is the growth promoting effect of applying *B. subtilis* SR22 on tomato plants, even if they are infected. This result is in agreement with that reported by Gohil et al. [33] for *Bacillus* sp. PG-8 on the growth of groundnut plants. The GC-MS analysis in this study revealed production of the bioactive compound 2,3-butanediol, which has been reported as a growth promoter for various plants [32]. This may explain the growth promoting effect of *B. subtilis* SR22 reported on tomato plants. Various growth promoting mechanisms have been discussed for *Bacillus* spp. including the production of growth regulators, siderophores, phosphate solubilization, nitrogen fixation, and the mitigation of abiotic and biotic stresses [34]. Improving the performance of the photosynthetic apparatus as well as the efficiency and content of their pigments has also been reported via enhancing the activity of the photosynthetic enzymes [35].

## 4. Materials and Methods

### 4.1. Tomato Cultivar and Used Microorganisms

Tomato seeds (cv. Castle Rock), obtained from Korma seed Co., Cairo, Egypt, were used in the greenhouse experiment. A virulent isolate of *R. solani* (AG-4 HG-I), originally isolated from rotted tomato roots, was obtained from the Plant Pathology Department, Faculty of Agriculture, Mansoura University, Egypt. The fungal pathogen was maintained on slants of potato dextrose agar (PDA, Difco, Detroit, MI, USA) at 4 °C until use. The endophytic bacteria were isolated from the stem of a faba bean plant that was cultivated in Kafr El-sheekh governorate, Egypt (30.98436° N, 31.29191° E). For inoculum preparation, the pathogen was cultured in a glass flask containing sterilized sorghum: soil medium (1:2, *v*/*v*) for 2 weeks at 28 ± 2 °C. 

To isolate the entophytic bacteria, plant tissues were cut into sections (1 mm^2^); the surface was sterilized using 90% ethanol for 3 min, followed by NaOCl solution (5%) for 3 min, rinsed in sterile water, and then aseptically plated on PDA plates and incubated at 28 °C for 2 days. The endophyte was picked up and transferred into PDA plates for purification. The purified colony was stained using Gram stain to examine the bacterial shape and Gram stain reaction, and designated SR22. For molecular identification, DNA was extracted using QiAamp DNA Mini Kit (Qiagen, Hilden, Germany). To amplify the 16S rDNA region, the primers 16S-27F: 5′-AGAGTTTGATCMTGGCTCAG-3′ and 16S-1492R:5′-CGGTTACCTTGTTACGACTT-3′ were used, following the method adopted by White et al. [36]. The nucleotide sequence was aligned and compared with the GenBank database via the NCBI search tool, BLAST. The phylogenetic tree of the obtained sequence and the closest BLAST sequences from the GenBank database were constructed based on the maximum likelihood method, using MEGA X software version 10.2.4. The inoculum of the endophyte was prepared by culturing on potato dextrose broth for 5 days at 30 °C. A spore suspension was adjusted at 10^5^ cfu mL^−1^ and supplemented with 3% gum arabic.

### 4.2. Antagonism Assay of B. subtilis SR22 against R. solani In Vitro

The antagonistic potential of *B. subtilis* SR22 was investigated against *R. solani* in vitro using the dual culture technique. In a PDA plate, *B. subtilis* SR22 was streaked 3 cm from the plate edge, and a 6 mm-diameter disc, taken from a 3-days-old culture of *R. solani*, was inoculated 2 cm from the opposite edge. A PDA plate only inoculated with a disc of *R. solani* was used as a control. For each treatment, 4 replicates were used. The plates were incubated at 28 °C and the test was ended when the control plate had a full growth. The growth inhibition (%) of *R. solani* was determined compared to the control plate.

### 4.3. GC-MS

The secondary metabolites of *B. subtilis* SR22 were identified using a GC-MS QP2010 system (Shimadzu, Kyodo City, Japan). The analysis was performed as follows: the detector mass spectrometer voltage was 75 eV at a max temperature of 260 °C. The capillary column (TRB-5MS, 30 m × 0.25 mm × 0.25 µm) was used. The GC program started at 60 °C for 3 min, and the oven temperature was increased at 5 °C min^−1^ until 260 °C where it was held for 10 min. The identity of the secondary metabolites was determined using the database of the National Institute of Standards and Technology (NIST 11) Spectral Library (Gaithersburg, MD, USA).

### 4.4. Greenhouse Experiment

Tomato seeds that were soaked in the spore suspension of *B. subtilis* SR22 for 5 h were sown in pots (20 cm diameter) and filled with sterilized soil (clay: sand, 1:2, *v*/*v*) at 5 seeds per pot. In addition, another dose (5 mL/seedling) of the spore suspension was applied as a soil drench 21 days post sowing. Tomato seeds soaked in sterilized water with 3% gum arabic were sown in another pot and served as the control treatment. Soil infestation was achieved by mixing the inoculum of *R. solani* with the upper layer of the soil at 2% (*v*/*v*), 14 days post planting. Five pots were used for each treatment. NPK fertilization was applied per kg of soil as follows: ammonium nitrate (350 mg), super phosphate (250 mg), and potassium sulfate (150 mg). All the pots were arranged in a complete randomized design, regularly watered (one time a week), and kept under greenhouse conditions at 31/25 °C day/night, 70% humidity. The applied treatments were as follows: C: non-treated with *B. subtilis* SR22 and uninfected; P: non-treated with *B. subtilis* SR22 and infected; B: treated with *B. subtilis* SR22 and uninfected; and P+B: treated with *B. subtilis* SR22 and infected.

#### 4.4.1. qRT-PCR

Seven days post-infection (dpi), the samples of the tomato roots from each treatment were collected for molecular investigation. The mRNA extraction of the samples was carried out using RNeasy Mini Kit (Qiagen, Hilden, Germany). The obtained mRNA was quantified using a NanoDrop 1000 spectrophotometer (Thermo Fisher Scientific Inc., Wilmington, DE, USA), and then stored at −20 °C. For cDNA synthesis, a reaction mixture with a total volume of 20 μL was used, containing mRNA (30 ng, 3 μL); dNTPs (10 mM, 2.5 μL); reaction buffer (2.5 μL); primer (5 pmol μL^−1^, 5 μL); reverse transcriptase enzyme (New England Biolabs, Germany) (0.2 μL); and RNase free water (6.8 μL). The reaction was carried out using a SureCycler 8800 thermocycler (Agilent, Santa Clara, CA, USA) at 42 °C for 1 h, and at 70 °C for 10 min, and then the product was stored at −80 °C. A qRT-PCR was performed using a Rotor Gene 6000 system (Qiagene Inc., Germantown, MD, USA). A reaction mixture with a total volume of 20 μL was used, containing cDNA (3 μL); SYBR Green Master Mix (Bioline, Germany) (12.5 μL); primer F + R (1.5 μL + 1.5 μL); and sterile RNase free water (1.5 μL). The sequences of the tested primers are presented in Table 6. β-actine was used as a reference gene. The real-time PCR program was carried out as follows: one cycle at 95 °C for 3 min, and 45 cycles at 95 °C for 15 s, 56 °C for 30 s, and 72 °C for 30 s. For each sample, three biological and three technical replicates were used. The comparative CT method (2^−ΔΔCT^) was used to analyze the relative expression level [37].

#### 4.4.2. Disease Assessment

Five tomato plants at forty-five dpi from each treatment were carefully uprooted and evaluated for disease incidence (DI) and severity (DS). For the DS assessment, the damage level in the tomato roots and hypocotyl was rated on a 5-degrees scale, as described by Carling et al. [38], where 0: no damage; 1: low discoloration; 2: discoloration and small necrosis; 3: discoloration and large necrosis; and 4: full death. This was calculated as follows:$$\mathrm{DS}\;\left(\%\right)=\frac{\sum^{\;}ab}{AB}\times 100$$
where *a* = number of rotted roots at the same degree, *b* = degree of disease, *A* = total number of assessed roots, and *B* = the highest degree of disease.

#### 4.4.3. Plant Growth Evaluation

Five tomato plants at forty-five dpi from each treatment were uprooted and evaluated for shoot and root lengths, shoot and root dry weights, and number of leaves. The dry weights were recorded after the samples were dried in a drying oven at 80 °C for 48 h.

#### 4.4.4. Estimation of Total Phenolic Content and Enzymes Activities

Forty-five dpi, total phenolic content, and enzymes activities were estimated in tomato roots of different treatments. For preparation of the crude extract, 3 g of fresh roots was ground in 5 mL phosphate buffer (100 mM, pH 7). The homogenate was centrifuged at 10,000 rpm for 15 min, the supernatant was then collected to serve as a crude extract in the next analyses. For each treatment, five replicates were applied.

The total phenolic content was estimated using the Folin–Ciocalteu method, as described by Malik and Singh [39]. The activity of the POD enzyme was estimated according to Maxwell and Bateman [40], and the activity of the PPO enzyme was estimated according to Galeazzi et al. [41].

### 4.5. Statistical Analyses

The obtained results were analyzed using the CoStat software (version 6.4, CoHort Software, Monterey, CA, USA) [42]. The data were first examined for normality and then subjected to an analysis of variance. The means were compared using Tukey’s HSD test at *p* ≤ 0.05 based on a one-way ANOVA [43].

## 5. Conclusions

This study reported a potent antagonistic activity of the endophyte *B. subtilis* SR22 against *R. solani* in vitro. Twenty secondary metabolites produced by *B. subtilis* SR22 were detected by a GC-MS analysis. These metabolites included various bioactive ones such as chlorogenic acid, pyrrolo [1,2-a]pyrazine-1,4-dione, hexahydro, propyl thioglycolic acid, phthalic acid, and 2,3-butanediol. The results from the greenhouse experiment revealed a considerable biocontrol activity for *B. subtilis* SR22 against Rhizoctonia root rot of tomato, recording a 53.8% reduction in the disease severity and incidence. Molecular investigation using a qRT-PCR showed a triggering effect for applying *B. subtilis* SR22 on the transcriptional expression level of the responsive factor *JERF3* and the defense-related genes *POD* and *PR1* in infected tomato plants. In addition, the total phenolic content and activity of the antioxidant enzymes POD and PPO in tomato roots were also induced due to *B. subtilis* SR22 treatment, indicating its resistance-inducing effect on tomato plants. Moreover, a growth promotion of tomato plants mediated by *B. subtilis* SR22 was also reported. In this regard, this treatment enhanced most of the evaluated growth parameters in tomato plants. The obtained results indicated that all aims of the study were achieved. Based on these results, we can conclude the promising role for *B. subtilis* SR22 as an immunity inducer and growth promoter in tomato plants against Rhizoctonia root rot, which may qualify it as an effective biocontrol agent. This biocontrol agent may benefit tomato-growers and may be efficient against this disease on others crops, at least from the same family. However, the field evaluation for this bio-agent is highly recommended singly and/or in association with other bio-agents for developing a highly effective and eco-safe biofungicide.

## Figures and Tables

**Figure 1 plants-11-02051-f001:**
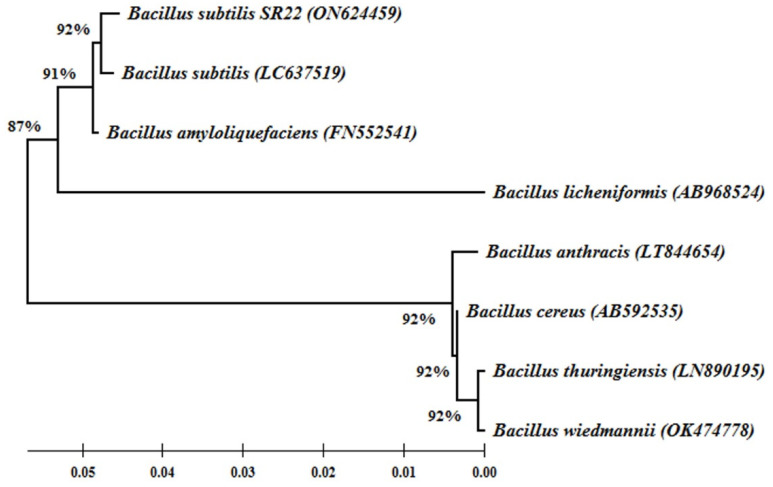
Phylogeny tree showing the relationship between the endophytic *Bacillus subtilis* SR22 and the closely related strains from the GenBank. Bootstrap values (%) are shown at the branches.

**Figure 2 plants-11-02051-f002:**
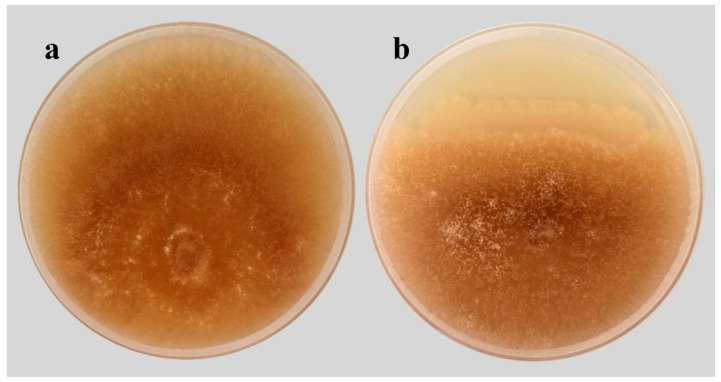
A dual culture test revealing the antifungal activity of *Bacillus subtilis* SR22 against *Rhizoctonia solani*, where (**a**) control plate, and (**b**) dual culture plate.

**Figure 3 plants-11-02051-f003:**
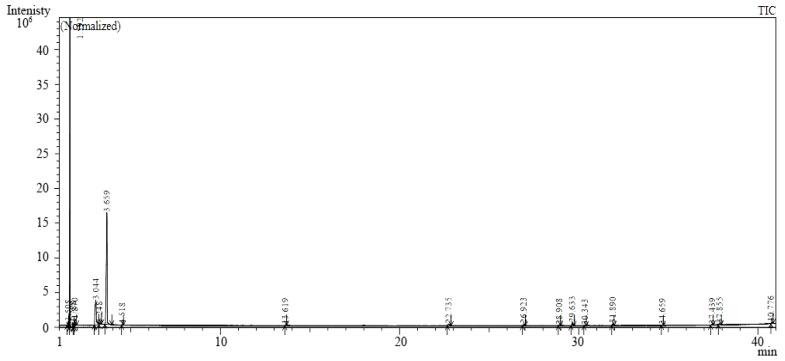
GC–MS chromatogram showing secondary metabolites produced by *Bacillus subtilis* SR22. Arrows show peak limits.

**Figure 4 plants-11-02051-f004:**
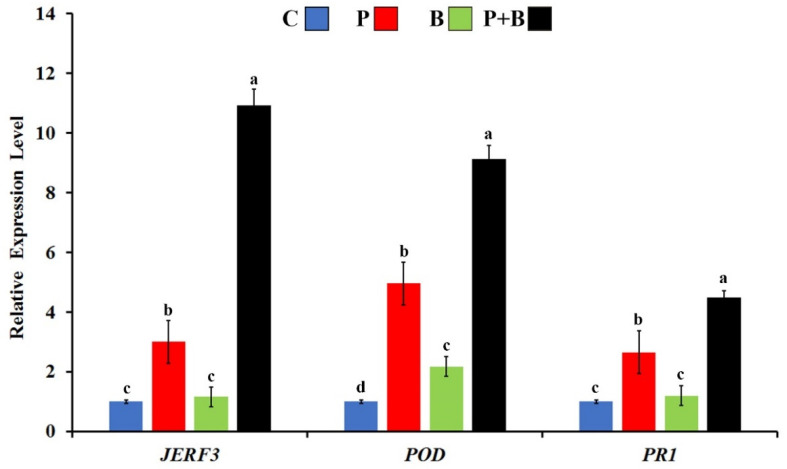
Expression profile of jasmonate and ethylene-responsive factor 3 (*JERF3*), peroxidase (*POD*), and pathogenesis-related protein 1 gene (*PR1*) in tomato roots infected with *Rhizoctonia solani* and/or treated with *Bacillus subtilis* SR22 at 7 dpi. C: non treated and uninfected; P: non-treated and infected; B: uninfected and treated with *B. subtilis* SR22; and P+B: infected with *R. solani* and treated with *B. subtilis* SR22. For each gene, columns superscripted with the same letter are not significantly different according to Tukey’s HSD test at *p* ≤ 0.05. For each sample, three biological and three technical replicates were used. Error bars represent standard errors.

**Figure 5 plants-11-02051-f005:**
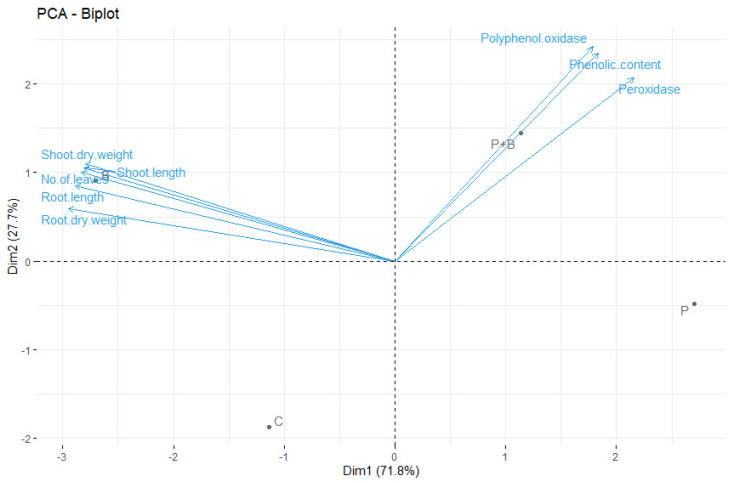
Ordination biplot of principal component analysis of the studied variables. C: non-treated with *B. subtilis* SR22 and uninfected; P: non-treated with *B. subtilis* SR22 and infected; B: treated with *B. subtilis* SR22 and uninfected; and P+B: treated with *B. subtilis* SR22 and infected.

**Table 1 plants-11-02051-t001:** Secondary metabolites of *Bacillus subtilis* SR22 as detected by GC-MS system.

Peak #	Retention Time (min)	Peak Area (%)	Compound Name
1	1.508	1.50	Propyl thioglycolic acid
2	1.592	33.87	Chlorogenic acid
3	1.788	1.09	2,3-Butanediol
4	1.870	1.90	1-Hexadecanol
5	3.044	13.70	Pyrrolo [1,2-a]pyrazine-1,4-dione, hexahydro
6	3.248	0.57	Acetamide
7	3.659	41.47	Propane, 2-ethoxy-
8	4.518	0.22	Propanamide
9	13.619	0.27	2-Piperidinone
10	22.735	0.89	3-Isopropoxy-1,1,1,7,7,7-hexamethyl-3,5,5-tri
11	26.923	0.61	Benzeneethanamine, N-[(pentafluorophenyl)
12	28.908	0.20	(S)-(+)-1,2-Propanediol
13	29.633	1.03	Phthalic acid, 3,5-dimethylphenyl 4-isopropyl
14	30.343	0.08	Phthalic acid, di(3,5-dimethylphenyl) ester
15	31.890	0.59	1,7-Di(3-ethylphenyl)-2,2,4,4,6,6-hexamethyl
16	34.659	0.13	1,2-Diphenyltetramethyldisilane
17	37.439	0.22	2,5-Piperazinedione, 3,6-bis(2-methylpropyl)
18	37.855	0.54	3-(2-N-Acetyl-N-methylaminoethyl)indol
19	40.776	0.35	Bis[di(trimethylsiloxy)phenylsiloxy]trimethyl
20	41.311	0.80	Heptasiloxane, hexadecamethyl-

**Table 2 plants-11-02051-t002:** Effect of *Bacillus subtilis* SR22 application on disease severity and incidence of tomato plants infected with Rhizoctonia root rot at 45 dpi *.

Treatment	Disease Incidence (%)	Disease Severity (%)	Severity Reduction (%)
C	0.0 ^c^	0.0 ^c^	0.0 ^b^
P	100.0 ^a^	65.0 ± 3.2 ^a^	0.0 ^b^
B	0.0 ^c^	0.0 ^c^	0.0 ^b^
P+B	60.3 ± 2.5 ^b^	30.0 ± 2.4 ^b^	53.8 ± 2.0 ^a^

* In each column, values followed by the same letter are not significantly different according to Tukey’s HSD test (*p* ≤ 0.05); each value represents the mean of three replicates ± SD. C: non-treated with *B. subtilis* SR22 and uninfected; P: non-treated with *B. subtilis* SR22 and infected; B: treated with *B. subtilis* SR22 and uninfected; and P+B: treated with *B. subtilis* SR22 and infected.

**Table 3 plants-11-02051-t003:** Mean growth parameters of tomato plants infected with Rhizoctonia root rot when treated with *Bacillus subtilis* SR22 at 45 dpi *.

Treatment	Shoot Length (cm)	Root Length (cm)	Shoot Dry Weight (g)	Root Dry Weight (g)	No. of Leaves
C	21.5 ± 2.1 ^b^	13.5 ± 0.9 ^b^	1.43 ± 0.07 ^b^	0.33 ± 0.04 ^b^	7.7 ± 0.6 ^a^
P	15.7 ± 3.0 ^c^	10.1 ± 0.7 ^c^	0.88 ± 0.09 ^d^	0.24 ± 0.03 ^c^	7.3 ± 0.5 ^a^
B	29.1 ± 1.5 ^a^	16.8 ± 1.0 ^a^	1.97 ± 0.10 ^a^	0.39 ± 0.03 ^a^	8.0 ± 0.7 ^a^
P+B	20.8 ± 1.8 ^b^	12.7 ± 0.9 ^b^	1.18 ± 0.06 ^c^	0.30 ± 0.05 ^b^	7.6 ± 0.3 ^a^

* In each column, values followed by the same letter are not significantly different according to Tukey’s HSD test (*p* ≤ 0.05); each value represents the mean of three replicates ± SD. C: non-treated with *B. subtilis* SR22 and uninfected; P: non-treated with *B. subtilis* SR22 and infected; B: treated with *B. subtilis* SR22 and uninfected; and P+B: treated with *B. subtilis* SR22 and infected.

**Table 4 plants-11-02051-t004:** Mean phenolic content and activity of peroxidase (POD) and polyphenol oxidase (PPO) enzymes in roots of tomato plants infected with Rhizoctonia root rot and treated with *Bacillus subtilis* SR22 at 45 dpi *.

Treatment	Phenolic Content (mg.g^−1^ Fresh wt)	POD (∆A_470_ min^−1^ g^−1^ Fresh wt)	PPO (∆A_420_ min^−1^ g^−1^ Fresh wt)
C	125.3 ± 2.2 ^d^	1.419 ± 0.07 ^d^	1.374 ± 0.06 ^d^
P	290.7 ± 2.7 ^b^	2.993 ± 0.06 ^b^	1.984 ± 0.04 ^b^
B	221.5 ± 3.1 ^c^	2.215 ± 0.04 ^c^	1.775 ± 0.09 ^c^
P+B	387.0 ± 5.4 ^a^	3.157 ± 0.08 ^a^	2.204 ± 0.08 ^a^

* In each column, values followed by the same letter are not significantly different according to Tukey’s HSD test (*p* ≤ 0.05); each value represents the mean of three replicates ± SD. C: non-treated with *B. subtilis* SR22 and uninfected; P: non-treated with *B. subtilis* SR22 and infected; B: treated with *B. subtilis* SR22 and uninfected; and P+B: treated with *B. subtilis* SR22 and infected.

**Table 5 plants-11-02051-t005:** Pearson correlation coefficient (r) between the tested variables.

	Shoot Length	Root Length	Shoot Dry Weight	Root Dry Weight	Number of Leaves	Phenolic Content	Peroxidase	Polyphenol Oxidase
Shoot length	1							
Root length	0.91 ***	1						
Shoot dry weight	0.90 ***	0.82 **	1					
Root dry weight	0.72 **	0.62 *	0.73 **	1				
Number of leaves	0.35 ^ns^	0.44 ^ns^	0.52 ^ns^	0.38 ^ns^	1			
Phenolic content	−0.28 ^ns^	−0.31 ^ns^	−0.25 ^ns^	−0.38 ^ns^	−0.16 ^ns^	1		
Peroxidase	−0.38 ^ns^	−0.41 ^ns^	−0.37 ^ns^	−0.50 ^ns^	−0.24 ^ns^	0.96 ***	1	
Polyphenol oxidase	−0.22 ^ns^	−0.27 ^ns^	−0.18 ^ns^	−0.36 ^ns^	−0.17 ^ns^	0.98 ***	0.97 ***	1

* = significant at *p* ≤ 0.05; ** = significant at *p* ≤ 0.01; *** = significant at *p* ≤ 0.001; while, ns = not significant.

**Table 6 plants-11-02051-t006:** Primer sequences used in the molecular investigation.

Gene Description	Abbrev.	Accession No.	Sequence (5′-3′)
Jasmonate and ethylene-responsive factor 3	*JERF3*-F *JERF3*-R	AY383630	GCCATTTGCCTTCTCTGCTTC GCAGCAGCATCCTTGTCTGA
Peroxidase	*POD*-F *POD*-R	X94943	CCTTGTTGGTGGGCACACAA GGCCACCAGTGGAGTTGAAA
Pathogenesis-related protein 1	*PR1*-F *PR1*-R	M69247	ACTTGGCATCCCGAGCACAA CTCGGACACCCACAATTGCA
β-actin	*β-actin-*F *β-actin* -R		GTGGGCCGCTCTAGGCACCAA
			CTCTTTGATGTCACGCACGATTTC

## Data Availability

Not applicable.

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
