# Peer review of "Endophytic Bacillus subtilis SR22 Triggers Defense Responses in Tomato against Rhizoctonia Root Rot"

_plants, 2022, doi:10.3390/plants11152051_

Round 1
Reviewer 1 Report
The current study entitled“Endophytic Bacillus subtilis SR22 mediates defense-responses elicitation against Rhizoctonia root rot and growth promotion in tomato” is good. For a better understanding in-depth, it is a need time to work on this topic. Furthermore, achieving potential benefits by using current technology depends on extensive research work for more exploration. Although the experiment is well organized, I suggest a major revision due to the following deficiencies.
Abstract
- Make the title a simple statement.
- Give the problem statement in a single line.
- Give a reason for the selection of the current technique. i.e., Endophytic Bacillus subtilis. What is a novelty in your work when this technology is already adequate for elicitation against Rhizoctonia root rot and growth promotion in tomatoes? Please clear that.
- Quantitative data is also essential to support your conclusion. Would you please provide quantitative data regarding the percentage significant increase or decrease in the abstract?
- Please provide a definitive conclusion withdrawn through research in a single line.
- Give future prospective in a single line.
- As per standard suggestions, please avoid using title words as keywords.
Introduction
- Please follow the title and rewrite the introduction in the following sequence, i.e., Endophytic Bacillus subtilis SR22, Rhizoctonia root rot, growth promotion in tomato, problem statement, aims of the study, and hypothesis.
- Also, provide a novelty statement at the end. What new things have authors done or correlated in this research compared to old ones?
- Would you please give a single line about the knowledge gap your research has covered along with the hypothesis statement?
Material and methods:
- No information is provided for isolation soil. Please give the GPS location of sampled plants from where isolation was done.
- Before sowing what seeds, treatment was done. No information is provided in terms of reference.
- Give details of fertilizers and irrigation schedule.
- Please provide a reference for statistical analysis.
- Please provide details of the homogeneity of data. If authors have not done that, then give a reason.
- Give details of software developer, country, etc. Also, provide its reference.
Results and Discussion.
- The results are very descriptive. Please give only significant results.
- I request the authors to provide a Pearson correlation of studied attributes.
- Also, provide principal component analysis or parallel plots for a better and easy understanding of data.
- Each table must have all details of abbreviations used in the table.
- Also, give mechanistic discussion. It is not a correct way to discuss results based on other scientists’ findings. Please elaborate on specified mechanisms that are regulating and result. Please rewrite the results and discussion again.
Conclusion
- Add the targeted beneficiary audience who will get benefits from this research.
- No lines are provided showing whether either author has achieved their aims of study or not.
- Also, give clear-cut recommendations while describing the best treatment.
Author Response
Thank you for your time and effort you have put into this review. We have mostly incorporated the suggested changes. Please find attached the detailed responses to your comments.
Best Regards

Reviewer 2 Report
In my opinion, the manuscript submitted for review contains important data from the point of view of a phytopathologist, which has application potential. The manuscript is well and carefully prepared in terms of content and graphics. I have no detailed comments on this work, therefore I believe that it can be accepted for publication in its current form.
Author Response
Thank you for your time and effort you have put into this review.
Round 2
Reviewer 1 Report
Dear Editor
I am satisfied with all the corrections incorporated in the manuscript. After fine correction in the English language paper can be accepted.
Author Response
The manuscript has been revised for English language. We hope the revised manuscript is satisfactory for reconsideration for publication in Plants.